# The fluidic memristor as a collective phenomenon in elastohydrodynamic networks

Alejandro Martínez-Calvo [1,2,9], Matthew D. Biviano [3,9], Anneline H. Christensen [3], Eleni Katifori [4,5], Kaare H. Jensen [3] & Miguel Ruiz-García [6,7,8]

Fluid flow networks are ubiquitous and can be found in a broad range of contexts, from human-made systems such as water supply networks to living systems like animal and plant vasculature. In many cases, the elements forming these networks exhibit a highly non-linear pressure-flow relationship. Although we understand how these elements work individually, their collective behavior remains poorly understood. In this work, we combine experiments, theory, and numerical simulations to understand the main mechanisms underlying the collective behavior of soft flow networks with elements that exhibit negative differential resistance. Strikingly, our theoretical analysis and experiments reveal that a minimal network of nonlinear resistors, which we have termed a 'fluidic memristor', displays history-dependent resistance. This new class of element can be understood as a collection of hysteresis loops that allows this fluidic system to store information, and it can be directly used as a tunable resistor in fluidic setups. Our results provide insights that can inform other applications of fluid flow networks in soft materials science, biomedical settings, and soft robotics, and may also motivate new understanding of the flow networks involved in animal and plant physiology.

Fluid flow networks—interconnected structures of elements that transport liquids or gasses—can be found in a wide range of both human-made and living systems, including oil pipelines, water supply systems, and the vasculature of animals and plants[1–3]. These systems have been usually modeled as collections of linear elements, i.e. resistors that obey Ohm's law. This approach is simple and powerful, allowing for easy assessment of flow and pressure distribution within the system[4]. It also facilitates the study of network properties, such as robustness, hierarchy, and phenomena like network remodeling and tuning[5–11]. However, experiments have shown that natural flow networks can contain intrinsically nonlinear elements that do not conform to Ohm's law, such as valves or vessels within the circulatory systems of plants and animals that respond to changes in pressure by varying the hydraulic resistance[12–15]. Flow networks containing these nonlinear elements are not well understood, and linear models are insufficient for capturing their behavior.

We begin by considering two biological cases in which the resistance of the network element is fundamentally nonlinear (Figs. 1a, b). In particular, Fig. 1a shows a schematic of the flow network corresponding to mammalian brain vasculature. When a cerebral arteriole, a small

[1]Princeton Center for Theoretical Science, Princeton University, Princeton, NJ 08544, USA. [2]Department of Chemical and Biological Engineering, Princeton University, Princeton, NJ, USA. [3]Department of Physics, Technical University of Denmark, DK 2800 Kgs. Lyngby, Denmark. [4]Department of Physics and Astronomy, University of Pennsylvania, Philadelphia, PA 19104, USA. [5]Center for Computational Biology, Flatiron Institute, New York, NY 10010, USA. [6]Departamento de Estructura de la Materia, Física Térmica y Electrónica, Universidad Complutense Madrid, 28040 Madrid, Spain. [7]GISC - Grupo Interdisciplinar de Sistemas Complejos, Universidad Complutense Madrid, 28040 Madrid, Spain. [8]Department of Mathematics, Universidad Carlos III de Madrid, 28911 Leganés, Spain. [9]These authors contributed equally: Alejandro Martínez-Calvo, Matthew D. Biviano. ✉e-mail: miguel.ruiz.garcia@ucm.es

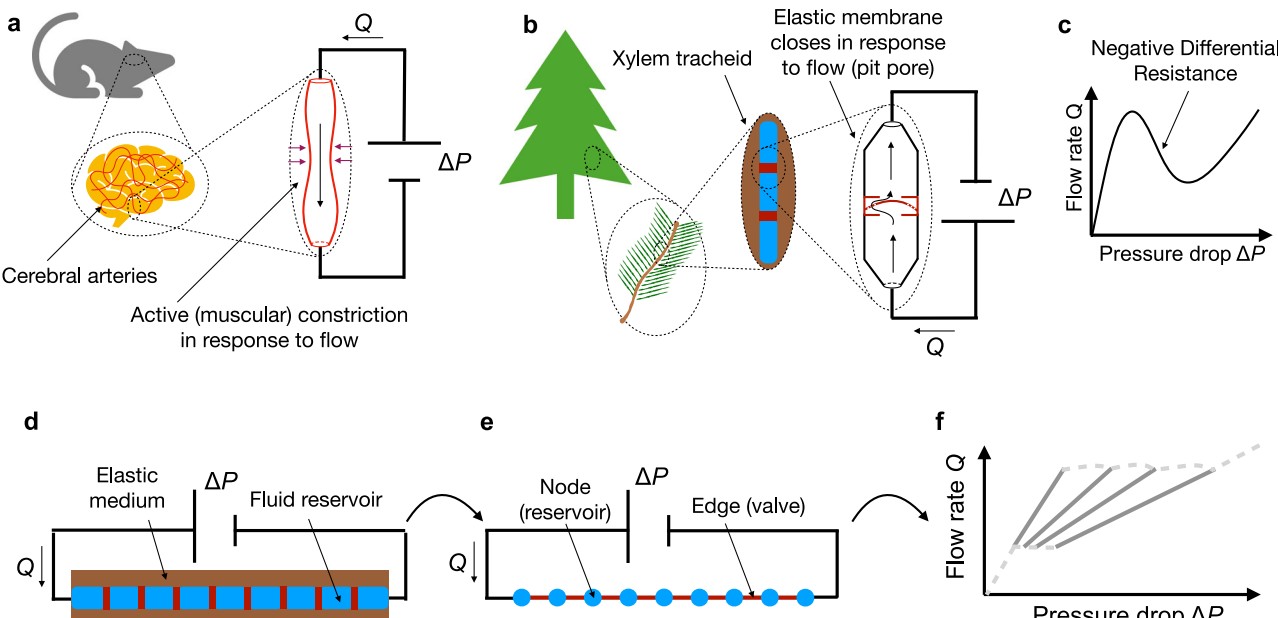

**Fig. 1 | Negative differential resistance is ubiquitous across biological systems.** **a**, **b** Examples of systems in which the relationship between flow rate and pressure difference exhibits a region of negative differential resistance, i.e., a region of negative slope, as schematized in (**c**). Because of this nonlinear behavior, such flow elements are referred to as nonlinear resistors. **a** Schematic of mammalian brain arteries, in which the muscles covering the blood vessels respond to flow by either expanding or contracting. When a vessel is extracted, isolated, and connected to a pump, the flow rate as a function of pressure difference displays a behavior qualitatively similar to that shown in (**c**)[12,13]. **b** Schematic of pit pores in xylem cells of trees from the gymnosperm family. The flow-rate-pressure-difference relation in **c** emerges from the passive mechanism of a porous membrane that responds to pressure difference by closing the flow channel[14]. **d** Schematic of our model system, in which many nonlinear resistors are connected in series to an external pump that controls the global pressure difference applied to the system. **e** displays a flow network analogue to (**d**), where nodes and edges are identified with volume reservoirs and valves, respectively. **f** Flow-rate versus pressure-difference relationship displayed by systems such as (**d**) and (**e**). The relationship is multivalued, and each line can be followed depending on the pressure protocol applied. Since the lines have different slopes, the global resistance displayed by the system has memory.

artery, is isolated from the animal, connected to an external pump, and the flow is measured, the response to the imposed pressure difference is qualitatively similar to the nonlinear curve shown in Fig. 1c, which exhibits a region of negative slope–also known as *negative differential resistance* (NDR). This effect is controlled by the muscles surrounding the artery that can expand or contract the vessel[12,13]. Figure 1b displays a schematic of the Gymnosperm (e.g., conifers and cycads) plant vasculature. In this case, the pit pores separating the xylem tracheids have an elastic membrane that can close depending on the flow rate[14], thus giving rise to a nonlinear flow rate response with a region of negative differential resistance similar to Fig. 1c. In the case of the arteries, a plausible explanation for this behavior is that they have evolved to ensure that the blood flow going into an organ does not exceed safe levels, actively responding to pressure[16] and flow[12,17,18]. Similarly, pit pores in the xylem tracheids may have evolved to ensure that the flow of sap stays within adequate levels and that they increase resistance to embolism (cavitation)[19,20]. However, in both cases, arteries and pit pores are interconnected in a flow network comprising many of these elements, motivating the need to understand the collective phenomena emerging in systems of NDR resistors.

It has long been known that systems exhibiting negative differential resistance often display instabilities and a wide variety of complex behaviors. Semiconductors that present NDR in bulk or when forming heterostructures can display heterogeneous electric field distributions, hysteresis loops, or self-sustained oscillations[21–25]. Electrochemical systems are also a paradigmatic case of nonlinear phenomena and can exhibit NDR and a broad range of complex nonlinear phenomena[26–29]. However, while most electric systems have their fluidic counterparts, the study of flow networks of NDR elements has not been explored to the same extent.

Artificial and biologically-inspired microfluidic networks are rapidly evolving to incorporate nonlinear elements and more complex topologies[30–42], including several examples of artificial valves, some of which exhibit NDR[14,33,35,36,42–48]. Although connecting these nonlinear valves in fluid networks could be straightforward, we will show that complex phenomena emerges when: (i) the system is able to locally store volume and (ii) the local volume changes are coupled to the pressure distribution along the system. These conditions are already present in the biological systems discussed above, in which volume accumulation inside the vessels/cells compresses the external medium surrounding the network (see Fig. 1d), but they have yet to be included in an engineered device. Our results shed light on the fundamental principles underlying complex phenomena in networks of nonlinear resistors, advancing the understanding of both biological and non-biological systems. We show that it is feasible to build these complex systems in the laboratory and harness their collective phenomenology. Altogether, our work helps establish a framework to predict and control emergent phenomena in networks of NDR elements, opening new avenues for harnessing such complex phenomena in the laboratory.

## The fluidic memristor
In this section, we present theoretical results, realistic simulations, and experiments of a 1D flow network of NDR elements. Our phenomenological model explains the fundamental mechanisms that lead to the emergent complex phenomena. In a nutshell, if the network was composed of linear resistors, the pressure distribution along it would decay homogeneously, leading to the same pressure drop at each resistor, and a global resistance that would be the sum of the individual resistors connected in series. However, when several NDR elements are connected in series and the spaces between them can accumulate

volume, the flow along the system in response to an externally imposed pressure (i.e. a pump) displays complex behavior and memory effects, see Fig. 1d and f. The reason behind this behavior is that there is a critical pressure above which a homogeneous pressure decay along the system is unstable. When that threshold is surpassed, the system divides into two regions, low and high-pressure drop. As the pressure keeps increasing the system displays consecutive hysteresis loops that lead to a global resistance that depends on the history of the protocol applied, a global resistance with memory. A feature reminiscent of the electric memristor[49], and a possible new paradigm for soft-matter systems that present memory effects[50–52].

## Phenomenological model: domain formation and nested hysteresis loops

We analyze here the qualitative behavior of fluid networks consisting of NDR resistors, as shown in the sketch in Fig. 1d. We use a phenomenological model, which is detailed in the Methods section and illustrated in Fig. 1e. In this model, each edge represents a NDR valve, and each node represents the region in between valves, which can accumulate volume due to the elasticity of the walls (as shown in Fig. 1d). To illustrate the phenomenology of this system, we are considering a 1D network of 30 nodes and 29 edges (systems comprising different numbers of nodes show equivalent behavior, as discussed in

following sections). We consider pressure-driven flow, and thus control the inlet pressure of the system, $P_{inlet}$, which is the pressure at the left-hand side of the 1D network (as shown in Fig. 1d, e). The outlet pressure is constant at $P_{inlet} = 0$. The two edges adjacent to the inlet and outlet follow a linear (Ohmic) relationship, which mimics an experimental setup in which solid tubes are connected to both ends of the fluid network. The other 27 internal edges of the network follow a nonlinear pressure-flow relationship with a region of negative differential resistance, as shown in Fig. 2a—the mathematical expression corresponding to this model is provided in the supplementary information (SI). It is important to emphasize that the behavior of an isolated NDR resistor (as displayed in Fig. 2a) is completely stable, meaning that the same curve is followed independently of the pressure protocol. However, a minimalist system containing one NDR valve connected to regions that can accumulate volume and other linear resistors can display bistability, see the supplementary information for more details.

We perform time-dependent numerical simulations with our 1D network of 27 NDR resistors by imposing the inlet pressure protocol described in the inset of Fig. 2b, where the colors red and blue indicate pressure ramping up and down, respectively. The inlet pressure at the contact is varied slowly, i.e., quasi-statically, so the network quickly adapts and does not display any dynamical behavior over time. The

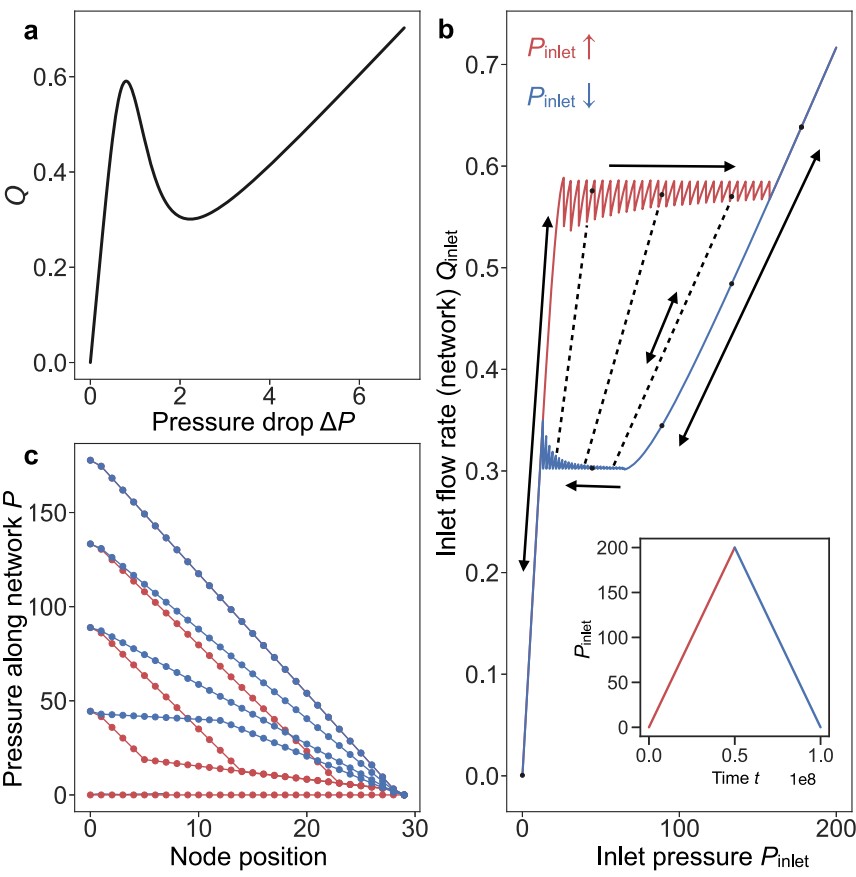

**Fig. 2 | Domain formation and hysteresis loops in flow networks of NDR elements. a** The graph shows the flow rate $Q$ as a function of pressure difference $\Delta P$ for a single nonlinear resistor. **b** Inlet flow rate $Q_{inlet}$ is plotted as a function of inlet pressure $P_{inlet}$ for a 1D network consisting of 30 nodes (27 nonlinear valves and 2 linear valves at the extremes of the network, see sketch in Fig. 1e). The inlet pressure is increased and decreased linearly as shown in the inset, while the outlet pressure is kept at zero. Nonlinear valves follow the black curve in (**a**). We set $h = 0.1$ for the linear valves, and $\alpha = 0.001$ (see Methods). The flow-rate versus pressure curve exhibits a hysteresis loop in which the upper and lower flow rate branches display 27 jumps, each jump corresponding to the position at which one nonlinear

resistor has swapped its state. Arrows in (**b**) indicate which side of the hysteresis loop can be followed in one or two directions, and dashed lines show three examples of other inner hysteresis loops. **c** The pressure $P$ is plotted as a function of position in the 1D network at different points along the hysteresis loop, indicated with dots in (**b**). Different slopes correspond to low and high-pressure drop domains, corresponding to regions in which nonlinear resistors are at the first or second positive differential resistance branch in (**a**). The colors follow the criteria displayed in the inset to (**b**). For a detailed explanation of the phenomenon please refer to the SI.

inlet and outlet flow rates are identical and plotted as a function of the pressure drop in Fig. 2b. The curve in Fig. 2b displays a clear hysteresis loop, in which the upper and lower branches of the loop display small jumps. Each jump corresponds to another inner hysteresis loop, as shown by the three dashed lines. Indeed, there are 26 inner stable branches in this system. Single- and double-arrow lines indicate the branches that can be followed in one or both directions by varying the global pressure drop, respectively. In particular, the upper and lower branches can only be followed by increasing and decreasing $P_{\text{inlet}}$, respectively, while left, right, and inner branches can be followed in both directions. For instance, if the system is in the upper red branch and we start decreasing $P_{\text{inlet}}$, the network will descend the closest inner stable branch. Fig. 2c displays the pressure distribution inside the system at different positions along the hysteresis loop, denoted with black dots in Fig. 2b. Note that when two domains are formed in Fig. 2c (low and high-pressure drop), the system can display one of two symmetric configurations: low-pressure drop first or high-pressure drop first. In Fig. 2c we show a simulation where the system finds one of the solutions going up and the complementary going down. In this work, we do not explore how boundary conditions, history of the protocol or other factors affect the appearance of one or the other solution.

We find that the network behaves as a tunable resistor, in which the global resistance depends on the history of the protocol applied. This leads to two fascinating phenomena: (i) The upper and bottom branches of the hysteresis loop present an approximately constant flow independently of the pressure applied to the system, equivalent to a zero differential resistance. (ii) The same pressure difference applied to the network, $P_{\text{inlet}}$, leads to multiple possible flow configurations, corresponding to the different branches that are selected depending on the protocol followed. Moreover, the inner branches have different slopes corresponding to different differential resistance. Thus, this memory effect controls the effective resistance of the

system, which led us to term the system a *fluidic memristor*. A more detailed explanation of the intricacies of this phenomenon is provided in the Methods section. Our goal now is to prove the existence of these effects in realistic fluid dynamical systems, opening a new avenue to harness such emergent phenomena in experimental setups.

## Numerical simulations of realistic elastohydrodynamic networks with NDR valves

To test whether our minimal phenomenological 1D model can predict the behavior of realistic systems, we first perform full-time-dependent numerical simulations of a 1D network consisting of two-dimensional (2D) valves (see the schematic in Fig. 3a and Materials and Methods). In particular, we consider two elastic and nearly incompressible blocks clamped at their outer boundaries, creating a channel through which fluid can flow due to an applied pressure difference $\Delta P$ between the inlet and outlet. The valves inside this channel are connected in series and are composed of two elastic rods clamped at the elastic blocks, which exhibit similar steady-state behavior to the curve in Fig. 2a, i.e., they exhibit a region of negative differential resistance (see Supplementary Information for more details). The deformation of the clamped elastic blocks allows the space between the valves to swell or shrink, and it couples volume accumulation with the pressure field. For simplicity, we also assume that inertia is negligible, implying that viscous effects are dominant. Such an approximation is valid if the Reynolds number comparing inertial and viscous effects is small, which can be achieved in small systems conveying small flow rates of viscous fluids. Under this approximation, the elastic rods and blocks deform quasi-statically due to pressure and viscous stresses.

We perform time-dependent numerical simulations of an array of 8 nonlinear valves. Both sides of the channel are connected to narrower channels that act as linear resistors, equivalent to the two linear resistors used in our phenomenological 1D model. We control the pressure at the inlet, $P_{\text{inlet}}$, i.e. at the left narrow channel, and impose a

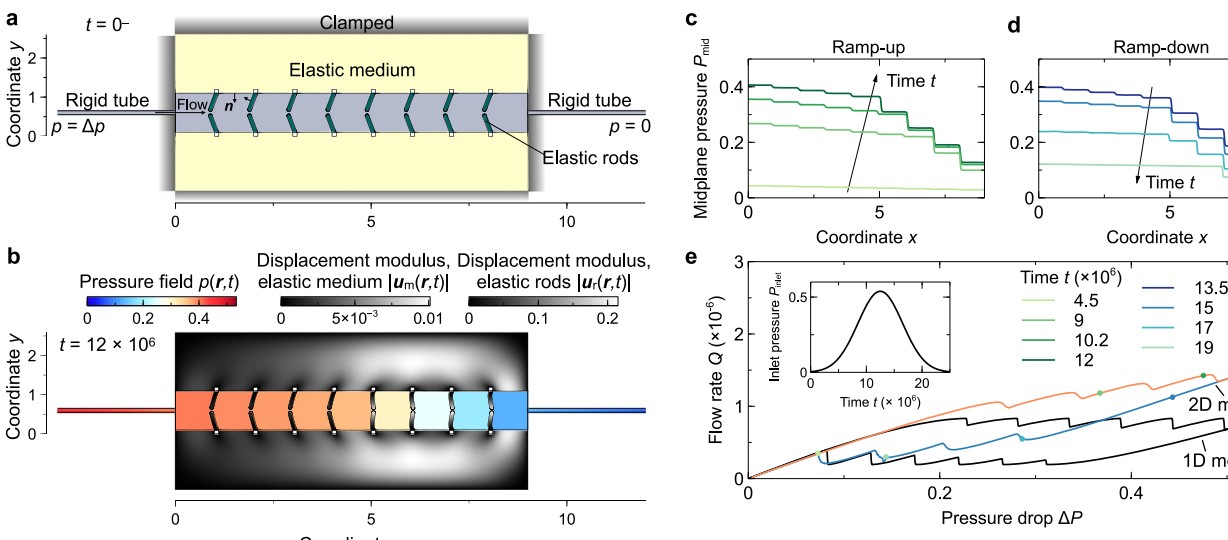

**Fig. 3 | Numerical simulations of a realistic flow network with NDR elements.** **a** Schematic of a 2D realistic system with 8 valves. The system consists of two elastic blocks clamped at their outer boundaries, which form a channel where the fluid flow passes through. The valves are located inside this channel and are composed of two elastic rods clamped at the elastic blocks. The contact interface between the fluid and the elastic rods and blocks is a free deformable surface. Two long rigid and narrow channels are positioned at the inlet and outlet of the channel, where we apply a pressure difference $\Delta P$. **b** Displays color plots of the flow pressure field $p(\mathbf{x}, t)$, and the elastic blocks and rods displacement field $|\mathbf{u}_m(\mathbf{x}, t)|$ and $|\mathbf{u}_r(\mathbf{x}, t)|$, respectively, at a dimensionless time of $t = 12 \times 10^6$. **c** and **d** depict the midplane pressure $P_{\text{mid}}$ as a function of $x$ position at different times, when the pressure is

monotonically increased in (**c**), and when it is monotonically decreased in (**d**), following the protocol for the inlet pressure $P_{\text{inlet}}$ shown in the inset to (**e**). For low $P_{\text{inlet}}$, the pressure drop along the system is homogeneous until there is spontaneous pattern formation, in which the system divides into low and high pressure drop domains. **e** shows the flow rate $Q$ as a function of the pressure difference $\Delta P$ following the inlet pressure protocol displayed by the inset. The flow-rate-pressure curve displays a large hysteresis loop. Dots of different colors correspond to the pressure profiles at the times shown in (**c**, **d**). Each jump exhibited by the hysteresis loop corresponds to one valve swapping its configuration as explained by the phenomenological model.

zero pressure at the outlet, as in the 1D model. Fig. 3e shows the inlet flow rate $Q_{inlet}$ as a function of the applied pressure difference, $\Delta P = P_{inlet}$, for the protocol described in the inset, equivalent to that of the 1D model. As the pressure is varied quasi-statically, the inlet and outlet flow rates are identical, i.e. $Q_{inlet} = Q_{inlet} = Q$. Fig. 3c and d show the pressure field along the midplane of the system at different times, corresponding to the dots in Fig. 3e. As predicted by our 1D model, the hysteresis cycle is connected to the creation of two domains of low and high-pressure drop, corresponding to the two slopes present in the lines shown in Fig. 3c and d. These two domains are produced by two groups of valves, each one operating in a different positive differential resistance branch. The upper line in Fig. 3d represents the pressure along the middle line of Fig. 3b, where the two domains are distinctly visible: the four valves which are closer to the inlet present almost no deformation (low-pressure drop), while the other four valves are highly deformed (high-pressure drop). As we found with the 1D model, the upper and lower branches of the hysteresis loop present small jumps that correspond to each of the valves swapping its configuration, thereby changing the size of the domains in Fig. 3c, d. In the 2D simulation protocol, we decrease the inlet pressure before all the valves swap to the second possitive differential resistance branch, as observed in the high-pressure curves of Fig. 3c, d. Thus, the ramping-down path of the hysteresis loop corresponds to one of the stable inner branches predicted by the phenomenological model.

Although we find a reasonable agreement between the numerical simulations and our phenomenological model, there is one quantitative difference: the upper and lower branches of the hysteresis loop do not correspond to a constant flow rate with superimposed jumps, but rather exhibit a monotonic increase in flow rate as the value of $P_{inlet}$ increases. To visualize this difference quantitatively, we include the prediction of the 1D model for a system of 8 valves. This difference stems from the fact that the 2D valves are composed of two compressible elastic rods that change their geometry due to the absolute

pressure, thereby affecting their response to $\Delta P$. To allow the phenomenological model to capture this behavior, we can use nonlinear resistors in which the flow not only depends on the pressure difference but also on the absolute pressure. Including this effect greatly improves the quantitative match between the phenomenological model and the realistic simulations, as presented in section 5 of the supplementary information.

## Building the fluidic memristor in the laboratory

To demonstrate the feasibility of constructing the fluidic memristor and employing it in real fluidic systems, we build an experimental platform based on the physical principles described in the previous sections. Specifically, we employ valves that exhibit a region of negative differential resistance, whose design draws inspiration from the operating principle of pit pore valves that are present in the xylem cells of many gymnosperm trees[53,54]. The behavior of pit-pore valves has been studied extensively in the past[14,55–57]. As illustrated in Fig. 4a, our valves consist of an elastic membrane stretched over a narrow gap of thickness ∼ 0.3 mm, with a pore in the center that enables fluid flow through the valve. As the pressure difference across the valve increases, the elastic membrane deforms, eventually blocking the pore. We connect this valve in parallel with a short tube of polytetrafluoroethylene (PTFE) with a very small inner diameter, which acts as a high-resistance bypass for the valve (see Fig. 4a). This structure enables complete control of the flow response to the pressure difference. We design the valves to close at a pressure of 40 mBar, approximately. Therefore, this system exhibits the flow versus pressure-difference relationship shown in Fig. 4c.

To replicate the behavior observed in our theoretical model and numerical simulations, we constructed an experimental flow network consisting of an array of 8 valves, which is contained in a flexible PDMS tube constrained from the outside by a rigid acrylic tube (see details in Methods section), as shown in Fig. 4a. The PDMS tube acts as the

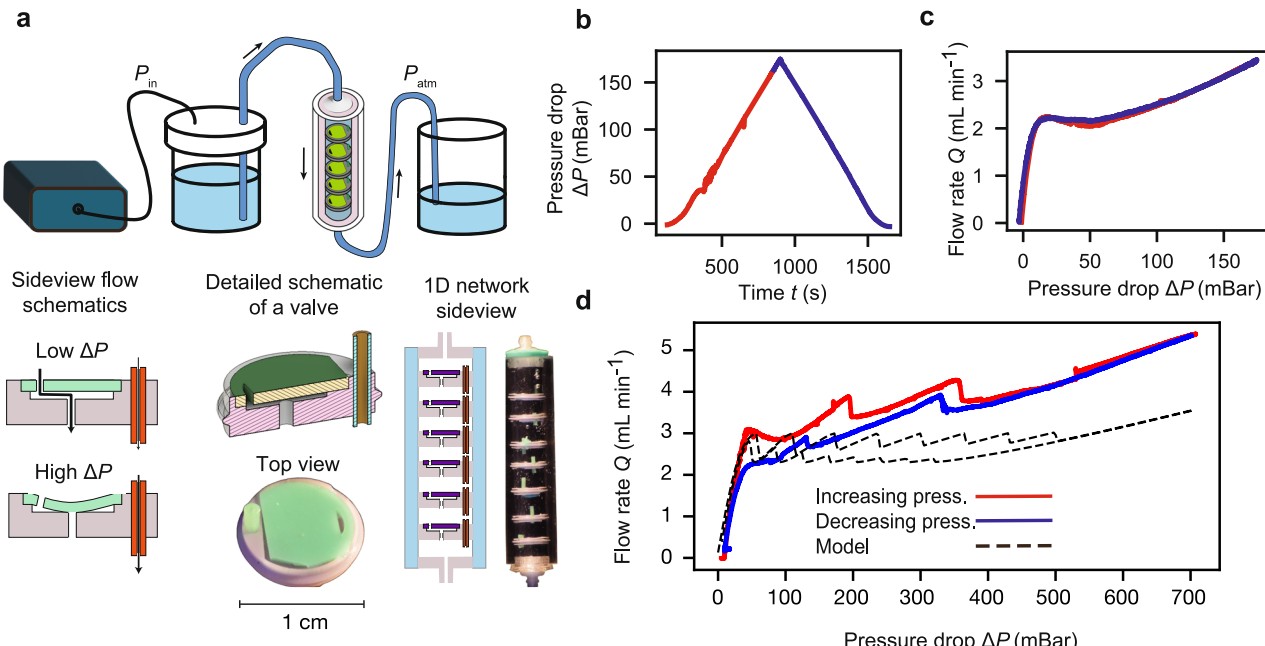

**Fig. 4 | Experiments on a flow network of NDR elements. a** Experimental setup. At the top, there is a schematic of the setup, while the bottom left shows a schematic of a single valve used to construct the experimental 1D flow network. The bottom right shows a schematic of the experimental network containing NDR valves in series inside an elastic tube with a rigid external layer. **b** depicts the protocol for the inlet pressure $P_{inlet}$, in which the pressure is monotonically increased and then decreased over time, as in the 1D model and full numerical simulations. **c** Flow rate as a function of the pressure difference for one valve. **d** Flow rate as a function of the pressure difference for the 8-valve experimental network. The network exhibits hysteretic behavior as predicted by the theoretical model and full numerical simulations. The jumps are a consequence of the valves swapping from one PDR branch to another. The dashed curve corresponds to the solution obtained from the phenomenological model.

elastic medium with clamped boundary conditions used in the realistic simulations. In our experiments, we gradually increase the inlet pressure from 0 to 800 mBar and then decrease it back to 0 mBar over a period of 8 hours, as depicted in Fig. 4b. We measure the resulting flow rate $Q$ and find a clear hysteresis loop, displaying jumps that indicate valves swapping from one positive differential resistance branch to another, similar to our phenomenological model and 2D numerical simulations (see Methods).

To compare the experimental results with the theoretical model, we use the functional form of the experimental valve shown in Fig. 4a to simulate the phenomenological 1D model (see SI). The results obtained from the 1D model are shown with dashed lines in Fig. 4d. The 1D model captures the order of magnitude of the resulting flow rate and the size of the hysteresis loop in pressure difference. Moreover, the experimental shape of the flow-rate-pressure-difference curve also qualitatively agrees with the model. However, the experimental data is tilted upwards, as observed in the 2D simulations. We attribute this behavior to valves changing their response to the pressure difference depending on the absolute pressure, which is not included in the phenomenological model. Additionally, we find that the experimental flow network exhibits a smaller number of jumps than the total number of valves. We suspect that imperfections in the valves cause them to not swap one by one but in groups, resulting in a smaller number of jumps.

## Similar behavior in other systems and impact on our work

Despite the relevance of fluid flow networks of nonlinear resistors in nature, the physics and emergent phenomena of such systems have remained poorly understood. To address this knowledge gap, we have combined experiments, theory, and numerical simulations to unravel the main mechanisms underlying the collective behavior of soft flow networks with NDR elements. Strikingly, our work reveals that a minimal network of NDR elements in series, which we have termed as a 'fluidic memristor', exhibits history-dependent resistance, allowing this minimal system to store information. Additionally, we demonstrate that it is feasible to build such a system in the laboratory and harness its collective phenomenology.

Similar hysteretic behaviors to the one described in this work can be found in other systems of strikingly different nature. For instance, in Lithium batteries, the relationship between chemical potential and ion concentration in each of the particles forming the electrode displays a non-monotonic behavior, leading to a hysteresis curve when considering a system of many particles[58,59]. Within soft matter physics, instabilities in rubber balloons and soft actuators have practical applications in the development of soft robotics[60–64]. In general, rubber balloons present a non-monotonic pressure-volume relationship[65–67]. In the context of cylindrical balloons, the non-monotonic relationship results in phase coexistence, where instability leads to the formation of two distinct regions, each having a different radius. The Maxwell construction can be applied to this process, which is characterized by the absence of hysteresis[68]. However, when multiple spherical balloons are connected to a common source, they exhibit an instability that causes them to transition individually between two equilibrium states. This results in nested hysteresis loops, which are strikingly similar to those observed in our case[59,69]. However, there are key differences with our system. In our case, the variables are flow and pressure difference instead of chemical potential (pressure) and ion concentration (volume) for batteries (and balloons). Additionally, in the semiconductor and electrochemical realms[24–26,29] or in some biological systems[70–75], other examples also present non-monotonic relations that lead to complex collective phenomena. Until now, this phenomenology had not been carefully analyzed in the fluidic realm. Here we have shown that a fluid flow network of NDR resistors displays nontrivial phenomena, such as pattern formation

and hysteretic behavior. These features make the system suitable for its use as a resistor with a global resistance that depends on the memory of the applied protocol. Moreover, we have also shown how to build fluid flow networks of nonlinear resistors and how to harness their emergent phenomena in the laboratory.

The fluidic memristor is ready to be used in fluidic setups, where it can work as a tunable resistor whose resistance is controlled by the history of the pressure difference applied to it. We hope that our work will inspire further exploration of the complex phenomena that networks of nonlinear resistors can offer and pave the way for the development of new devices in the fluidic realm. Additionally, we believe that our work can motivate experimental studies in biological systems to investigate the role of these collective effects in networks of nonlinear resistors, such as animal and plant vasculature. Taken together, our experimental approach, theoretical framework, and findings provide a foundation for these promising future avenues for research.

## Methods
### Phenomenological model

To explore the response of flow networks of nonlinear resistors, we build upon previous work and use the mathematical model introduced in ref. 76. This previous work focused on the emergent dynamical behavior (self-sustained oscillations) present in this model and was purely theoretical, it did not explore the phenomenology explored here: multiple hysteresis loops and memory effects, neither it tried to build or simulate realistic examples of this type of networks. This model is constituted by phenomenological expressions that approximate the behavior of viscous fluid (-0 Reynolds number) flowing within a channel of flexible walls and going through several nonlinear valves. The model uses a flow network, a set of $N$ nodes and connections—the edges—between them, see Fig. 1e. Pressure, $P_i$, and accumulated volume, $V_i$, are defined at each node $i$ and can be time dependent. The volumetric current $Q_{ij}$ is defined as the current from node $i$ to node $j$ on edge $ij$. To establish the input and output of the network, in this work, we choose the two nodes at the extremes of the network and externally control the pressure, analogous to connecting a battery or pump to a resistor network.

The current between nodes $Q_{ij}$ depends on the pressure difference $\Delta P_{ij} = P_i - P_j$. In a simple Ohmic resistor, this relationship would be linear, but here we consider the following general pressure-flow relation:

$$Q_{ij} = \begin{cases} \frac{1}{2} V_i^{\beta} \Gamma(\Delta P_{ij}), & \text{if } P_i > P_j, \\ \frac{1}{2} V_j^{\beta} \Gamma(\Delta P_{ij}), & \text{if } P_i < P_j, \end{cases} \tag{1}$$

where $\Gamma(\Delta P)$ is a function that can be either linear in the pressure drop, $\Gamma_L(\Delta P) = 1/R_0 \Delta P$ (where $R_0$ is a dimensionless parameter), or nonlinear, $\Gamma_{NL}(\Delta P)$. Specific functional forms of $\Gamma_{NL}$ are detailed in the SI. The current also depends on the accumulated volume at the node from where it flows, where the exponent $\beta$ determines its functional dependence. In the extreme case where $P_i > P_j$ and $V_i = 0$, $Q_{ij}$ should be zero, because node $i$ is "empty" and there is no volume to flow from node $i$ to $j$. For semiconductor networks, $\beta = 1$[24,25], whereas for the fluid networks we consider here, the scaling of viscous flow leads us to choose $\beta = 2$. We note that for $\beta \in [0.5, 2]$, the dynamics are relatively insensitive to the choice of $\beta$, see[76].

Proceeding with our analysis, we consider mass conservation which determines the temporal variation of the accumulated volume at every node,

$$\frac{dV_i}{dt} = \sum_j -Q_{ij}. \tag{2}$$

Our sign convention assigns a positive sign to the current that leaves node $i$ when $\Delta P_{ij} > 0$. Finally, we assume that the walls of the channel are deformable and volume can change in the space between the valves (the nodes), see Figs. 1d and e. To include this effect in the model we add a phenomenological constitutive relationship between the excess volume from a baseline ($V_0$), and the pressure field:

$$V_i - V_0 = \alpha \sum_j L_{ij} P_j, \qquad (3)$$

where $L_{ij}$ is the $ij$ element of the graph Laplacian $L = D - A$, $D$ being the degree matrix defined as $D_{ij} = d_i \delta_{ij}$, with $d_i$ the degree of node $i$, $A$ the adjacency matrix, and $\delta_{ij}$ the Kronecker delta[77]. Without loss of generality, we set $V_0 = 1$. In semiconductors, Eq. (3) corresponds to the network Poisson's equation, which couples charge accumulation to the electric field. In a fluid network, it models the effect of a channel surrounded by elastic walls (Fig. 1d). If an elastic medium surrounds the flow network, volume accumulation inside a network node will be coupled to the pressure field via the deformation of the external medium. The numerical value of the coupling constant $\alpha$ depends on material properties and the detailed geometry. It is, in principle, possible to predict the value of $\alpha$ from analytical theories[78–80], or from direct numerical simulations. As shown in the main text, however, the qualitative behavior shown in this work is relatively insensitive to the value of $\alpha$, we therefore focus on the basic properties of the model and treat $\alpha$ as a fitting parameter. For the simulations of the phenomenological model we use $\alpha = 0.001$ to be in the stationary and small volume accumulation regime[76]. The final step in the model development involves combining the mass conservation and volume accumulation equations (2) and (3) into a single expression:

$$\alpha \sum_j L_{ij} \frac{dP_j}{dt} = -\sum_j Q_{ij}, \qquad (4)$$

which is the system of time-dependent ordinary nonlinear differential equations that we solve numerically, together with boundary conditions imposing the overall pressure drop.

## Understanding the intricacies of the nested hysteresis loops

To understand the mechanisms underlying the phenomena that leads to the nested hysteresis loops, let us divide the domain of the nonlinear function in Fig. 2a into three parts that we will denote as: first positive differential resistance (PDR) branch (from $\Delta P = 0$ to the local maximum), NDR branch (negative slope region), and second PDR branch (from the local minimum onward).

In the protocol displayed by the inset of Fig. 2b, we start by increasing $P_{inlet}$ from zero, represented by the red dots at the bottom of panel c (where each point indicates the pressure at each node of the network). As we increase $P_{inlet}$, all the nonlinear resistors of the network follow the black solid curve in Fig. 5, represented by the red dots on the first PDR branch of the curve, until now the pressure distribution is homogeneous inside the network. However, once all the resistors reach the local maximum of Fig. 5, if $P_{inlet}$ keeps increasing, a homogeneous pressure drop along the network becomes unstable (see SI). Beyond this pressure, the system finds a different stable solution by swapping one resistor to the second PDR branch of Fig. 5. Through this mechanism, as $P_{inlet}$ increases, one by one the nonlinear resistors jump to the second PDR branch, and the system divides into two distinct domains characterized by a high and a low pressure drop, as shown by Fig. 2c (red curves). Once all the resistors have swapped to the second PDR branch, the system displays an internal pressure drop that is homogeneous, as shown by the blue upper line in Fig. 2c. When the inlet pressure $P_{inlet}$ is decreased following the ramping-down protocol in the inset to Fig. 2b, represented by blue dots in Fig. 5, all the resistors stay on the second PDR branch until they reach the local

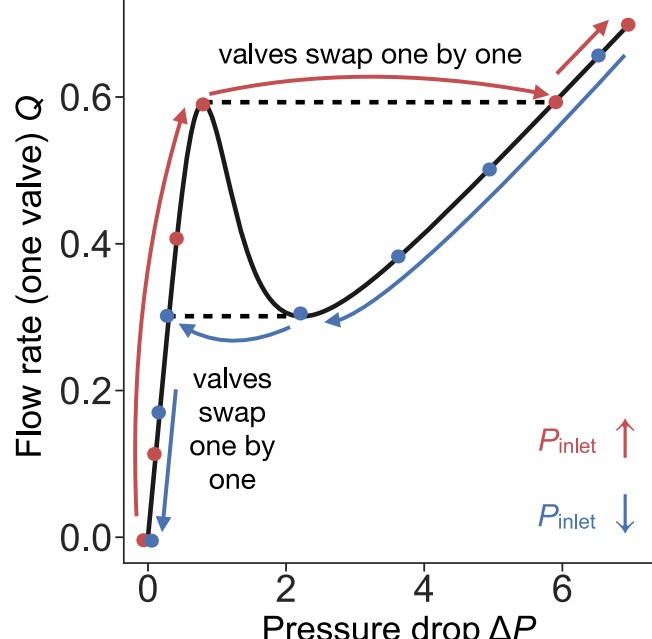

**Fig. 5 | Flow rate $Q$ as a function of pressure difference $\Delta P$ for a single NDR resistor.** Points and arrows explain the process that leads to the nested hysteresis loops presented in Fig. 2 for a 1D network of NDR resistors. NDR resistors swap one by one from one PDR branch to another as the total pressure imposed to the system ($P_{inlet}$) increases and then decreases.

minimum. Below such $P_{inlet}$, now one by one resistors jump to the first PDR branch, as shown in Fig. 5, again forming two pressure drop domains (bottom blue curve in Fig. 2c).

All the intricacies of the multiple hysteresis loops present in panel Fig. 2b can be understood using similar arguments to the ones used here. The jumps in the upper branch of the hysteresis loop are present because an infinitesimal increase in $P_{inlet}$ leads to a resistor jumping from the first to the second PDR branch. Since the pressure drop is externally controlled for the complete system ($P_{inlet}$), the pressure drop across the other NDR valves has to decrease to accommodate this sudden change, leading to the jumps present in Fig. 2b every time one NDR valve changes branch. A similar mechanism explains the different slopes for the stable branches in the nested hysteresis loops. We have a system of multiple valves connected in series, and since the valves are always in one of their two PDR branches, the effective resistance of the complete system is the sum of the individual resistances. With the peculiarity that now the valves can be in one of their two PDR branches (that have different resistances). Following this idea, the slopes of the left and right sides of the loop in Fig. 2b are directly proportional to the slopes of both PDR branches in Fig. 2a. However, the slope of the inner hysteresis branches (dashed lines in Fig. 2b) have slopes that interpolate between both limit values, depending on how many resistors are in each PDR branch. Finally, it is important to mention that the shape and properties of the multiple hysteresis loops shown in Fig. 2b can be directly controlled changing the shape of the nonlinear curve in Fig. 2a, and also by changing the number of resistors in the network.

## Continuum modeling and numerical simulations

To test whether our minimal network 1D model is able to capture the dynamics of a realistic system, we consider the fluid-elastic network schematized in Fig. 3. The system is comprised of two elastic materials with several elastic leaflets clamped at their inner boundaries, sandwiching a 2D Newtonian flow. We perform time-dependent numerical simulations coupling Stokes flow of an incompressible Newtonian

fluid, and linear elasticity for the elastic domains. To reduce the number of parameters, we non-dimensionalize the set of equations upon choosing the following spatial, displacement, velocity, pressure, time, and flow rate characteristic scales, respectively

$$\ell_c = u_c = h_g, \quad v_c = \frac{h_g G_r}{\mu}, \quad p_c = G_r,$$

$$t_c = \frac{\mu}{G_r}, \quad Q_c = \frac{G_r h_g^3}{\mu}, \tag{5}$$

where $h_g$ is the gap of the fluid region sandwiched between the two outer elastic media, $\mu$ is the viscosity of the liquid, and $G_r$ is the shear elastic modulus of the elastic rods. Hence, the fluid velocity field $\boldsymbol{v}(\boldsymbol{x}, t)$ and pressure $p(\boldsymbol{x}, t)$ are governed by the dimensionless continuity and momentum equations, which read:

$$\nabla \cdot \boldsymbol{v} = 0, \quad \text{and} \quad \boldsymbol{0} = \nabla \cdot \boldsymbol{T}, \tag{6}$$

where we have assumed that fluid inertia is negligible, thus $Re = \rho h_g v_c / \mu = \rho h_g^2 G_r / \mu^2 \ll 1$, $Re$ being the Reynolds number and $\rho$ the density of the fluid. Here $\boldsymbol{T} = -p\boldsymbol{I} + \nabla \boldsymbol{v} + \nabla \boldsymbol{v}^T$ is the fluid stress tensor. At the inlet, we impose the value of the pressure, and zero stress, which in dimensionless variables read

$$p = \beta \quad \text{and} \quad \boldsymbol{T} \cdot \boldsymbol{e}_x = \boldsymbol{0} \quad \text{at} \quad x = 0, \tag{7}$$

where $\beta = \Delta P / G_r$ is a compliance parameter measuring the ratio between the inlet pressure $\Delta P$ and the rod's elastic shear modulus $G_r$, and $\boldsymbol{e}_x$ is the stream-wise unit vector. As displayed in Fig. 3, the inlet pressure can be varied temporally, thus in that case $\beta = \beta(t)$ as $\Delta P = \Delta P(t)$. At the outlet we impose zero stress and we set the reference pressure to zero,

$$p = 0 \quad \text{and} \quad \boldsymbol{T} \cdot \boldsymbol{e}_x = \boldsymbol{0} \quad \text{at} \quad x = L. \tag{8}$$

At the contact surface between the liquid and the elastic media we impose continuity of velocities,

$$\boldsymbol{v} = \partial_t \boldsymbol{u}, \tag{9}$$

where $\boldsymbol{u}(\boldsymbol{x}, t)$ is the displacement field of the elastic materials. The elastic media satisfy the Cauchy equation of motion under the small-displacement approximation,

$$\boldsymbol{0} = \nabla \cdot \boldsymbol{\sigma}_i, \tag{10}$$

where we have also neglected elastic inertial effects, and the subscript $i = \{r, m\}$ denotes the *rods* and the *outer elastic matrix, respectively*. Here $\boldsymbol{\sigma}_i = 2\boldsymbol{\varepsilon}_i + \lambda_i / G_i \mathrm{tr}(\boldsymbol{\varepsilon}_i) \boldsymbol{I}$ is the solid stress tensor of a Hookean elastic material, $\boldsymbol{\varepsilon}$ is the strain tensor, and $G_i = E_i / [2(1 + v_i)]$ and $\lambda_i = E_i v_i / [(1 + v_i)(1 - 2v_i)]$ are the two Lamé constants, expressed in terms of the Young modulus $E_i$ and Poisson ratio $v_i$. We also consider the complete nonlinear expression of the strain tensor,

$$\boldsymbol{\varepsilon}_i = \frac{1}{2}(\nabla \boldsymbol{u} + \nabla \boldsymbol{u}^T + \nabla \boldsymbol{u} \cdot \nabla \boldsymbol{u}^T), \tag{11}$$

to accurately capture the deflection of the thin rods under the linear stress–strain relationship approximation, where $|\partial_j u_i| \ll 1$. Nevertheless, the results reported here are not significantly dependent on this geometric nonlinearity.

At the contact between the rigid shaft and the elastic material we impose clamping conditions, $\boldsymbol{u} = \boldsymbol{0}$. At the fluid–solid interfaces the

continuity of stresses must be fulfilled,

$$\boldsymbol{\sigma}_i \cdot \boldsymbol{n} = -\boldsymbol{T} \cdot \boldsymbol{n}, \tag{12}$$

where $\boldsymbol{n}$ is the unit normal vector to the liquid-solid interfaces.

The dimensionless parameters that govern the problem are the compliance parameter $\beta = \Delta P / G_r$, $\lambda_i / G_i$, the dimensionless rod's length $L_r / h_g$ and thickness $d_r / h_g$, the total dimensionless length of the system $L_m / h_g$, the thickness of the outer elastic medium, $d_m / h_g$, the medium-to-rod shear modulus ratio $G_m / G_r$, and the angle of the rod with respect to the vertical, $\theta$.

**Values of the dimensionless parameters used in simulations**

To obtain the results displayed by Fig. 3, the values of the dimensionless parameters for the leaflets are, $\lambda_r / G_r = 1/3$, $L_r / h_g = 0.48$, $d_r / h_g = 0.1$, $\theta = 20.75^o$, and $P_{inlet} = \beta(t)$, where $\beta(t)$ is the Gaussian function shown in the inset to Fig. 3e. The values of the dimensionless parameters for the outer elastic medium are, $\lambda_m / G_m = 10^3$ (nearly incompressible material), $G_m / G_r = 10$, $d_m / h_g = 1.5$, $L_m / h_g = 9$. The thickness and length of the solid inlet and outlet are set to 0.1 and 7, respectively, and the spacing between clamped leaflets is set to one.

## Valve Manufacturing and Multi-Valve Assembly

The valves were 3D printed in resin in ABS-like Photoresin (Elegoo, China) with an MSLA 3D printer (Sonic Mini 4K, Phrozen, Taiwan). Once printed, these valves were twice submerged in isopropyl alcohol (99%, Borup, Denmark) and sonicated for 10 minutes to remove excess resin. Once clean, these are then UV cured on both sides for 20 minutes in a UV chamber (CL-508-BL, Uvitec, UK). The rubber inserts were produced by the casting of a silicone based polymer (Elite Double 22, Zhermack, Italy). The molds for these castings are produced by the same method as used for the valves, but the curing is done for 1h on the mold side while submerged in 60C water. To assemble the singular valves the hole for the 1.59mm OD, 0.394mm ID PTFE tubing (IDEX, IL, USA) was drilled out with a handheld drill to 1.5mm and the tubing pressed into the resulting hole. The PTFE tubing is then cut to size with a razor blade. A diagram of these valves and their dimensions can be found in the supplementary information.

The experiment is housed in a two part tube, with an outer clear acrylic tube and an internal clear silicone tube (Sylgard 184, Dow Corning). The internal tube is made by first casting a sacrificial wax cylinder from the silicone mold (Elite Double 22, Zhermack, Italy) of a 12.5 mm diameter acrylic rod. The wax used was from a candle purchased at the local supermarket and heated till fully melted before pouring into a 50C preheated silicone mold. This wax rod is then placed inside a jig which concentrically centres both the wax rod and outer acrylic tube and the clear silicone is poured in and allowed to set for 2 days. Once fully cured, the wax is then melted out of the cylinder at 80C for 2–3 h in an oven and the cylinder is then washed in hot soapy water to remove any residual wax.

The end caps are produced by FDM 3D printing. They have been printed in black PLA (Ultimaker, Netherlands) on an Ultimaker 3+ (Ultimaker, Netherlands). They have been printed with 100% infill and also been a leak and fit checked before use.

To assemble the components, the valves are placed in the tube one by one and rotationally aligned for easy removal of bubbles. Once all the valves are in place, the end caps are added. All components were designed in Fusion 360 (Autodesk).

## Valve testing

The liquid used for all experiments is sucrose solution. The sucrose solution is produced by the dissolution of sugar (Dansukker, Denmark) in de-ionized water on a hotplate to achieve a 60% sucrose solution. This solution is then checked for viscosity on a desktop viscometer (NDJ-95, Vevor), and water is added to achieve the desired 22 mPa.s.

Solutions are used for 2 days before being remade and checked daily to ensure that they are compliant with the desired viscosity.

To construct the flow circuit we have a symmetrical flow system before and after the valves with a pressure sensor and flow sensor on both sides of valves. The pressure sensors used are 26PCCFG5G (Honeywell, NC, USA) that are connected to a HX711 based load cell amplifier (M5stack, China). These sensors are individually calibrated from a pressure sweep generated by a MFCS-EZ (Fluigent, France) across its working range (0-1Bar). The flow sensors used are SLF3S-1300F (Sensiron, Switzerland), and these are calibrated directly using a flow of our sucrose solutions originating from tank held 1 m above the tubing exit, where the mass is quantified over a period of a minute. These sensors are connected to an Arduino nano which prints the flowrate and pressure to an attached computer running a python script. To generate the pressure in the experiments, we have used the MFCS-EZ across a range of 0-0.85 Bar, to generate and control the pressure in the system. We use 4mm OD polyurethane tubing (Festo, Denmark) along with barbed luer lock connectors (Master-Flex,PA,USA) to join the sensors and valves. For the fluid reservoir, we have connected a pressure-pot type system where we use 4, 0.5L conical schott glass containers (212834454, Schott, Duran, Germany) with a draw tube to the bottom of the containers.

For the experiments we first gently fill the system and ensure the removal of all the bubbles from the system, typically by tilting the valve tube. Once removed, we run the full pressure sweep quickly (1h, 0-0.8-0 bar) to fully seat the valves, and then we run our long-term sweep. Typical long-term sweeps will run for 8 h from 0-0.8-0 bar in a triangular wave.

## Data availability
The data that support the findings of this study are available from the corresponding author upon request. Source data are provided with this paper.

## Code availability
Code reproducing the results of the phenomenological model is publicly available here: https://github.com/miguel-rg/fluidic_memristor.

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

## Acknowledgements

A.M.-C. acknowledges support from the Princeton Center for Theoretical Science and the Human Frontier Science Program through the grant LT000035/2021-C. M.R.-G. acknowledges support from the Ramón y Cajal program RYC2021-032055-I and from the CONEX-Plus program funded by Universidad Carlos III de Madrid and the European Union's Horizon 2020 research and innovation program under the Marie Skłodowska-Curie grant agreement No. 801538. E.K. acknowledges support from the UPenn MRSEC DMR-2309043, and the Simons Foundation through Investigator grant #568888.

## Author contributions

M.R.-G., E.K., and K.H.J designed research. M.R.-G. and E.K. developed the phenomenological model and M.R.-G. carried out the simulations. A.M.-C. carried out the realistic fluid dynamics simulations. M.D.B., A.C., and K.H.J. designed the experiments, and M.D.B. carried them out. All authors discussed the results and contributed to the final manuscript.

## Competing interests

The authors declare no competing interests.
