## [Peer Review File · Nature Communications]

The fluidic memristor as a collective phenomenon in elastohydrodynamic networksEditorial Note: Parts of this Peer Review File have been redacted as indicated to remove third-party material where no permission to publish could be obtained.

REVIEWER COMMENTS

Reviewer #1 (Remarks to the Author):

Networks with nonlinear electrical, mechanical, fluidic etc., elements are ubiquitous in both technology and the natural world. Motivated by examples of biological vasculature, the authors explore collective phenomena in a 1D (linear) elasto-hydrodynamic network by combining experiments, theory, and numerical simulations. A minimal phenomenological model for a soft, fluidic, nonlinear network is proposed and analyzed. The basic ingredient is the presence of a negative differential (hydraulic) resistance branch in the response of an individual element. The authors show that a minimal network composed of such nonlinear resistors can display memory effects and hysteretic behavior (hence act as a 'memristor'). A concrete realization using soft flow channels with flexible asymmetric valves is provided and the same qualitative phenomena are recapitulated using both numerical simulations with detailed fluid-structure interactions and an experimental demonstration. The work is quite interesting and well written. The analysis and results are largely easy to understand, but some points could be clearer still (detailed below). Overall, I am happy to recommend publication once the authors satisfactorily respond to my questions below.

My primary concern is with regard to the collective nature of the phenomena reported in this paper. In several places it is mentioned that an individual NDR is stable and not hysteretic, e.g., "the behavior of an isolated NDR resistor (as displayed in Fig. 2a) is completely stable, meaning that the same curve is followed independent of the pressure protocol." and in the discussion "a single valve is stable and instability only emerges as a collective phenomenon." This is a bit puzzling to me as Fig. 2a seems to show a hysteresis loop even for a single NDR element. Similarly in Fig. 4c, it is unclear if the single element in the experiment exhibits hysteresis or not (perhaps a zoomed in inset can be added?). The SI also only addresses a linear instability calculation for a 1D chain of NDRs, demonstrating a collective instability when operated on the descending branch of the flow curve. I couldn't figure out if and how the calculation can be used to demonstrate that a single NDR element is stable, perhaps this can be expanded on.

If a single element is stable, but the collective is unstable, then how many elements is required to be a collective? Is there a minimal number of the elements or valves for the 1D network to exhibit hysteresis and how does this depend on the flow curve properties of a single element? The phenomenological model and the detailed elasto-hydrodynamic simulations use different number of nodes and edges. How were the number of elements chosen and is there any size dependence in the response reported (except for the number of jumps)?

There is perhaps a useful opportunity here to contrast and compare the current proposal with similar but different phenomena in other soft systems. The hysteretic response with a spatial domain wall in the pressure gradient is strongly reminiscent of phase separation with the descending NDR branch corresponding to a spinodal region. As noted in the paper itself, this is also similar to the well known elastic Rayleigh-Plateau or Gent instability in rubber tubes, balloons, arterial aneurysms etc. (see for

instance Gent, A. N. "Elastic instabilities in rubber." *International journal of non-linear mechanics* 40.2-3 (2005): 165-175.; Mallock, A. "II. Note on the instability of India-rubber tubes and balloons when distended by fluid pressure." *Proceedings of the Royal Society of London* 49.296-301 (1891): 458-463). The latter has a proper mapping to phase separation and Cahn Hilliard theory (see recent work by John Biggins for example). Although the fluidic setup considered here is nominally similar to elastic instabilities mentioned, they are nonetheless distinct. In particular, elastic phase separation is driven by a negative static (differential) response (akin to negative elastic constant), which can occur in a passive system and it drives a thermodynamic instability. The fluidic NDR element instead has a negative dynamic (differential) response (akin to negative viscosity), which necessarily requires energy input/activity and can be used to produce work (is there a way to assess the energetics of pumping in the fluidic memristor?). The connection of a dynamic NDR in a single element and its (in)stability is less obvious though.

Another point is with regard to the choice of control variables. In Fig. 1c, the flow curve gives a unique flow rate for an imposed pressure drop, but if the flow rate is fixed, then the pressure drop exhibits multistability. So depending on what the control variable is, the system may or may not exhibit memory, hysteresis, multistability etc.

I think a better explanation and clarification of all these points would be very useful.

Fig. 3: how does the domain wall in the valve opening propagate? Presumably its speed is set by the ramping of P ? Do you ever see spontaneous drifting of the domain wall for strong enough static pressure drops, i.e., is there an analogue of a trigger (pushed) wave in the system?

The model also includes a capacitive element to account for local fluid storage. It was unclear to me what aspects of the phenomenology rely strongly on this feature, and what do not. Perhaps this can be explained a bit more.

In relating the single NDR element (Fig. 2a) to the collective (Fig.2b), is there a simple relation between the slopes of the response curves for the network in terms of the individual element flow properties?

Lines 399-401: "One plausible explanation for this difference stems from the fact that the 2D valves are composed of two compressible elastic rods that change their geometry due to the absolute pressure, thereby affecting their response to ΔP ." What happens if you keep ΔP the same and increase both the inlet and outlet pressure? This would be a simple check for this hypothesis.

Minor comments.

Eq. 3: The graph laplacian is currently unscreened corresponding to a Poisson equation. If one includes screening (so we get a Helmholtz-like equation), then there should be direct dependence on pressure itself. Can this explain the tilted response curves?

Eq. 7-8: What is e_{θ} ? Is the numerics performed in 2D or 3D? The longitudinal coordinate used in main text is ' x ' while the coordinate used in Method--simulations is a mixing of ' r ' (such as fluid

velocity field $v(r,t)$ and pressure $p(r,t)$ and ' \mathbf{x} ' (such as Eqs.7 and 8, and the displacement field of elastic materials). Also, both G and μ_s are used to denote the shear modulus. It would be useful if the notation were made common everywhere and explained more clearly.

Eq. 12: Shouldn't the boundary condition be continuity of the normal stress, so $\sigma_n = T_n$?

Reviewer #3 (Remarks to the Author):

This paper explores fluid flow networks that contain elements with a non-linear pressure-flow relationship, which is commonly observed in both biological and artificial systems. Through the integration of experimental data, simple theory, and numerical simulations, the authors provide an insight into the collective behavior of such networks. It is intriguing to consider the idea of a fluidic memristors, but the term is not entirely new (see Xiong et al. 2023), and the authors did not demonstrate many possible capabilities (e.g. calculating using such systems, or determining whether biological systems perform calculations). The paper is clear and well written (there were no typographical errors or problems with the presentation of the data). While the authors barely show the potential of this idea, the idea presented is very interesting, and I am in favor of accepting the work in its current form.

Reviewer #4 (Remarks to the Author):

The authors present an interesting and timely study on a "fluidic memristor", inspired by similar non-linear hysteretic memory-capable elements from electric circuits and beyond, but realised for the first time in the fluid flow network analogy. They combine theory, simulation, and experiments. They explore how this memristor operates, where the hysteresis originates from, and the coupling to elasticity in the chamber and valve elements. Ultimately, and unfortunately, though, the analytic model is illuminating for the effect but are a rather poor match to the simulation and especially the experiment, capturing the real behaviour only in the most qualitative, broad strokes. Finally, with a handful of exceptions detailed later, the paper is well-written, well-motivated, and easy to follow.

Taken all together, it is the opinion of this referee that the manuscript is promising enough to warrant eventual publication in Nature Communications, but requires further polish as well as at least one critical further step in the modelling approaches first.

The critical point first: other than the sawtooth-like behaviour of the flow rate as a function of the pressure drop, nothing about the 1d model matches the experiment, nor does it match particularly well to the 2d simulation. The authors even make a (strong!) speculation about a major reason for these mismatches -- the absolute pressure level is simply a fixed boundary condition in the model. However, it

would be both very relevant and should also be relatively easy to either make the absolute pressure level either a fitting parameter or at least a parameter to explore further. Given its claimed (and plausible) importance to rectifying the one glaring issue in the paper, this needs to be better addressed, at least in the 1d model. To be clear, the authors could also vary the absolute pressure in the simulation and/or the experiment and that would certainly also add value to the paper, but I think the phenomenological model would probably be enough.

Smaller points:

The breakdown of the protocol in figure 2 that begins on line 265 is too detailed, too confusing, and too much like a figure caption. Also, the "black solid curve in panel b" referred to in lines 268-9 does not appear to exist?

On line 322 "dots" in Fig. 3b are referred to and it is unclear what is meant by this.

The authors claim that there are two domains of behaviour "distinctly visible" in Fig. 3b. This is not so clear, and even it were clear, in order to better establish these two distinct domains, the authors should include some form of actual quantification to back up their assertion.

Generally, throughout all the figures, the authors use symbols, features, and line widths that are too small on a printed page. These should be enlarged to improve readability.

On line 481, in the discussion, the authors say "until now, this phenomenology had been elusive in the fluidic realm" which, in light of the examples of neurovasculature and gymnosperms discussed earlier in the paper and studied in detail elsewhere, I find to be a bit too strong of a statement.

In the methods section the authors seemingly swap notation from Q_{ij} to I_{ij} beginning in equation 2 with no warning or explanation.

Throughout page 14 and 15, the usage of G_r vs G is confusing and unclear. Also, it seems likely that G appears entirely mistakenly on line 646? That should likely be μ_s instead.

On line 632 the authors assert they can set the reference pressure to zero without loss of generality. This should be true in a linear system, but seems highly non-trivial here. The authors should either explain further or amend.

Reviewer #1:

Networks with nonlinear electrical, mechanical, fluidic etc., elements are ubiquitous in both technology and the natural world. Motivated by examples of biological vasculature, the authors explore collective phenomena in a 1D (linear) elasto-hydrodynamic network by combining experiments, theory, and numerical simulations. A minimal phenomenological model for a soft, fluidic, nonlinear network is proposed and analyzed. The basic ingredient is the presence of a negative differential (hydraulic) resistance branch in the response of an individual element. The authors show that a minimal network composed of such nonlinear resistors can display memory effects and hysteretic behavior (hence act as a 'memristor'). A concrete realization using soft flow channels with flexible asymmetric valves is provided and the same qualitative phenomena are recapitulated using both numerical simulations with detailed fluid-structure interactions and an experimental demonstration. The work is quite interesting and well written. The analysis and results are largely easy to understand, but some points could be clearer still (detailed below). Overall, I am happy to recommend publication once the authors satisfactorily respond to my questions below.

We thank Referee #1 very much for their positive opinion of our work. Indeed their comments have helped us very much to clarify a point that, we realize now, it was not clear in the previous version of the paper: the stability of one valve/the minimal system to observe hysteretic behavior. We are really thankful for helping us to improve our manuscript for the reader and hope that our paper is now ready to be accepted.

Questions:

Comment 1: My primary concern is with regard to the collective nature of the phenomena reported in this paper. In several places it is mentioned that an individual NDR is stable and not hysteretic, e.g., "the behavior of an isolated NDR resistor (as displayed in Fig.2a) is completely stable, meaning that the same curve is followed independent of the pressure protocol." and in the discussion "a single valve is stable and instability only emerges as a collective phenomenon." This is a bit puzzling to me as Fig. 2a seems to show a hysteresis loop even for a single NDR element. Similarly in Fig. 4c, it is unclear if the single element in the experiment exhibits hysteresis or not (perhaps a zoomed in inset can be added?). The SI also only addresses a linear instability calculation for a 1D chain of NDRs, demonstrating a collective instability when operated on the descending branch of the flow curve. I couldn't figure out if and how the calculation can be used to demonstrate that a single NDR element is stable, perhaps this can be expanded on. If a single element is stable, but the collective is unstable, then how many elements is required to be a collective? Is there a minimal number of the elements or valves for the 1D network to exhibit hysteresis and how does this depend on the flow curve properties of a single element? The phenomenological model and the detailed elasto-hydrodynamic simulations use different number of nodes and edges. How were the number of elements chosen and is there any size dependence in the response reported (except for the number of jumps)?

Response: Indeed, this was not clear enough in our previous version of the manuscript. We have modified Fig.2a in the main text to avoid confusing the reader with the arrows that explained the collective behavior of the system containing multiple NDR elements. Following the advice of Referee #4 this explanation is now in the Methods section, to avoid including too much information in the main text.

We have included a new detailed explanation in the supplementary information discussing the minimum ingredients that are necessary to see an unstable behavior for a system with only one NDR valve. In summary, whereas one completely isolated valve is stable, a system that allows for volume accumulation and include other resistors (they can be linear) can present an instability leading to a hysteretic behavior. We have copied here the complete explanation to facilitate the review of our manuscript:

“We discuss here the stability of a minimalist system where one valve with a region of negative differential resistance is connected to a pump through two linear resistors (rigid tubes). The pump imposes an external pressure difference to the complete system (ΔP). If the system cannot store volume anywhere, the valve will not present any instability. In other words, if we increase and then decrease ΔP the flow through the system does not present any bistability or hysteresis. This is because this situation does not allow any mismatch of the flows within the system.

However, if the walls of the tubes containing the fluid right before and after the valve can store volume, then the NDR region can be unstable:

Let us study the condition for which this system can be unstable. The pump imposes an external constraint for the pressure drops inside the system:

$$\Delta P = \Delta P_{NL} + 2\Delta P_L, \quad (1)$$

where ΔP_{NL} and ΔP_L are the pressure drops at the nonlinear resistor (valve) and linear resistors, respectively. Since there are two regions that can store volume due to elastic walls, see Fig. ??, mass conservation controls the volume in these two regions:

$$\begin{cases} \frac{dV_B}{dt} = \frac{\Delta P_L}{R} - Q(\Delta P_{NL}) \\ \frac{dV_A}{dt} = Q(\Delta P_{NL}) - \frac{\Delta P_L}{R} \end{cases} \rightarrow \frac{d(V_B - V_A)}{dt} = 2 \left(\frac{\Delta P_L}{R} - Q(\Delta P_{NL}) \right), \quad (2)$$

where V_B and V_A are the volumes in the regions before and after the valve, see Fig. ?. We also know that volume and pressure are coupled, in our model we consider elastic walls that couple volume with pressure as:

$$V_i = 1 - \alpha(P_{i+1} - 2P_i + P_{i-1}), \quad (3)$$

in this simple system, this leads to:

$$V_B - V_A = \alpha(-\Delta P + 3P_B - 3P_A) \rightarrow \frac{d(V_B - V_A)}{dt} \propto \frac{d(P_B - P_A)}{dt}, \quad (4)$$

it is clear that other simpler couplings such as $V_i \propto P_i$ will also lead to the same result. Combining now equations (2) and (4):

$$\frac{d(P_B - P_A)}{dt} \propto 2 \left(\frac{\Delta P_L}{R} - Q(\Delta P_{NL}) \right), \quad (5)$$

using now that $\Delta P_{NL} = (P_B - P_A)$ and equation (1), we get,

$$\frac{d\Delta P_{NL}}{dt} \propto 2 \left(\frac{\Delta P - \Delta P_{NL}}{2R} - Q(\Delta P_{NL}) \right). \quad (6)$$

Let's consider that the system is at a stationary solution with a pressure difference at the valve ΔP_{NL}^* that is within the region of negative differential resistance,

$$\frac{d\Delta P_{NL}^*}{dt} = 0, \quad (7)$$

then, we are interested in the evolution of a small perturbation,

$$\frac{d\Delta P_{NL}^* + \epsilon}{dt} \propto 2 \left(\frac{\Delta P - \Delta P_{NL}^* - \epsilon}{2R} - Q(\Delta P_{NL}^* + \epsilon) \right), \quad (8)$$

linearizing the flow through the valve around ΔP_{NL}^* we get,

$$\frac{d\epsilon}{dt} \propto -2 \left(\frac{1}{2R} + Q'(\Delta P_{NL}^*) \right) \epsilon, \quad (9)$$

where $Q'(\Delta P_{NL}^*)$ is the derivative of $Q(\Delta P_{NL})$ evaluated at ΔP_{NL}^* . Since ΔP_{NL}^* is in the NDR region, $Q'(\Delta P_{NL}^*) < 0$. The stationary solution will be unstable if,

$$\left(\frac{1}{2R} + Q'(\Delta P_{NL}^*) \right) < 0, \quad (10)$$

what leads to,

$$\frac{1}{2R} < -Q'(\Delta P_{NL}^*), \quad (11)$$

approximating the derivative in the NDR by a straight line going through the local maximum and minimum (see Fig. ??), we get,

$$\frac{1}{R} < \frac{2(Q_{max} - Q_{min})}{(\Delta P_{min} - \Delta P_{max})}. \quad (12)$$

To test this result we have carried out some simulations with $\alpha = 0.001$ (volumes will be close to one) and

$$\Gamma_{NL} = \Delta P \frac{1 + 0.1(\Delta P)^4}{1 + (\Delta P)^4}, \quad (13)$$

for this case (12) gives a stability threshold $\frac{1}{R} \approx 0.4$. Indeed when we carry out the simulations we see that linear resistors with a larger resistance than the threshold lead to bistability, whereas a system with linear resistors with smaller resistance do not show any unstable behavior:

This figure shows the numerical stability analysis for a NDR valve within a minimalist system containing two linear resistors and two regions that can accumulate volume (nodes) connected to an external pump. Instabilities lead to the bistable behavior of the system for $\frac{1}{R} < 0.4$, as the theory predicts. For each simulation we show three panels. Panels (a,d,g,j) display flow versus pressure difference for the nonlinear valve (continuous line) and the linear resistor (dashed line). Panels (b,e,h,k) show the flow through the complete system versus the external pressure imposed at the inlet by the pump, red and blue lines indicate increasing and decreasing pressure difference, respectively. Finally, panels (c,f,i,l) present the pressure in the four nodes of the system at different times (marked in panels (b,e,h,k) as black dots).

We have also validated this effect in the experimental setup. In this case a low linear resistor (R1) leads to a very small bistability region where a larger linear resistance (R2) leads to a larger bistability region as the theory predicts:

Comment 2: There is perhaps a useful opportunity here to contrast and compare the current proposal with similar but different phenomena in other soft systems. The hysteretic response with a spatial domain wall in the pressure gradient is strongly reminiscent of phase separation with the descending NDR branch corresponding to a spinodal region. As noted in the paper itself, this is also similar to the well known elastic Rayleigh-Plateau or Gent instability in rubber tubes, balloons, arterial aneurysms etc. (see for instance Gent, A. N. "Elastic instabilities in rubber." International journal of non-linear mechanics 40.2-3 (2005): 165-175.; Mallock, A. "II. Note on the instability of India-rubber tubes and balloons when distended by fluid pressure." Proceedings of the Royal Society of London

49.296-301 (1891): 458-463). The latter has a proper mapping to phase separation and Cahn Hilliard theory (see recent work by John Biggins for example). Although the fluidic setup considered here is nominally similar to elastic instabilities mentioned, they are nonetheless distinct. In particular, elastic phase separation is driven by a negative static (differential) response (akin to negative elastic constant), which can occur in a passive system and it drives a thermodynamic instability. The fluidic NDR element instead has a negative dynamic (differential) response (akin to negative viscosity), which necessarily requires energy input/activity and can be used to produce work (is there a way to assess the energetics of pumping in the fluidic memristor?). The connection of a dynamic NDR in a single element and its (in)stability is less obvious though.

Response: This is a great comment by Referee 1, which has helped us clarify this point in the paper. Let us discuss here the similarities and differences between the instabilities displayed by our system and the ones displayed by rubber balloons. Following one of the papers of John Biggins [1], one cylindrical rubber balloon can “phase separate” in two distinct regions to avoid an unfavorable homogeneous deformation. In particular following [1], consider a balloon that is locally stretched by a factor of λ , stores an elastic energy per unit length $w(\lambda)$ and hence bears a tension $T = \partial w / \partial \lambda$:

[REDACTED]

If we take one of these cylindrical balloons, and control the volume inside it, due to the concave region of the energy ($w(\lambda)$), the system can decrease the total energy creating two regions of different deformation (λ_a and λ_b). Analogously, this process can be understood looking at the tension, where the region of negative slope is unstable, and the system separates in the two regions corresponding to λ_a and λ_b . During this phase separation the system keeps a constant pressure (tension) that can be found through the equal area rule. This is similar to a evaporation/condensation phase transition that occurs at constant pressure as long as there is phase coexistence. Note that the coexistence pressure is the same when we are increasing or decreasing the volume of the system (there is no hysteresis).

It is indeed true that the flow versus pressure difference relationship of our fluidic valve is reminiscent of the tension (pressure) versus stretching (volume) relationship in the rubber balloon. However, our soft valve cannot be in two states at the same time, so one valve cannot “phase separate” as it is the case on one cylindrical balloon.

We think that our fluidic system may have a better analogue in spherical rubber balloons. Although the relation between internal pressure (tension) and volume still presents a region of negative slope, one isolated balloon does not present an unstable behavior, it cannot “phase separate” in two regions. If the volume inside the balloon is slowly decreased whereas the pressure is measured, one finds a smooth response as it is the case of our isolated fluidic valve, from [2]:

[REDACTED]

However, when N balloons are connected to each other, and one increases the volume in the system, all the balloons increase their volume together until they reach the negative slope, and now the system is globally unstable. As the total volume keeps increasing, one by one the balloons change from one stable branch to another. If one then decreases the total volume the balloons do the opposite and one by one jump from the second stable branch to the first one, from [2]:

[REDACTED]

We think that this behavior is very reminiscent of the behavior shown by our system of many NDR valves connected in series, although the magnitudes and nature of the system are of course completely different: pressure (flow) versus volume (pressure gradient) in the case of balloons (valves). Finally, please note how in the case of many spherical balloons there is no a coexistence pressure determined by the equal area rule, and the pressure displayed by the system when the total volume of the system is controlled depends on the protocol, as it occurs for the flow in our fluidic memristor. Following the advice of Referee #1 we have improved this discussion in the paper, and we have included the following new references for completeness [1 – 10].

[1] A. Giudici and J. S. Biggins. Ballooning, bulging, and necking: An exact solution for longitudinal phase separation in elastic systems near a critical point. *Physical Review E*, 102(3):033007, 2020.

[2] I. Müller and P. Strehlow. Rubber and rubber balloons: paradigms of thermodynamics, volume 637. Springer Science & Business Media, 2004.

- [3] A. Gent. Elastic instabilities in rubber. *International Journal of Non-Linear Mechanics*, 40(2-3):165–175, 2005.
- [4] T. J. Jones, T. Dupuis, E. Jambon-Puillet, J. Marthelot, and P.-T. Brun. Soft deployable structures via core-shell inflatables. *Physical Review Letters*, 130(12):128201, 2023.
- [5] A. Mallock. II. Note on the instability of india-rubber tubes and balloons when distended by fluid pressure. *Proceedings of the Royal Society of London*, 49(296-301):458–463, 1891.
- [6] J. T. Overvelde, T. Kloek, J. J. D’haen, and K. Bertoldi. Amplifying the response of soft actuators by harnessing snap-through instabilities. *Proceedings of the National Academy of Sciences*, 112(35):10863–10868, 2015
- [7] E. Chater and J. W. Hutchinson. On the propagation of bulges and buckles. 1984.
- [8] E. Ben-Haim, L. Salem, Y. Or, and A. D. Gat. Single-input control of multiple fluid-driven elastic actuators via interaction between bistability and viscosity. *Soft robotics*, 7(2):259–265, 2020.
- [9] B. Gorissen, D. Reynaerts, S. Konishi, K. Yoshida, J.-W. Kim, and M. De Volder. Elastic inflatable actuators for soft robotic applications. *Advanced Materials*, 29(43):1604977, 2017.
- [10] B. Van Raemdonck, E. Milana, M. De Volder, D. Reynaerts, and B. Gorissen. Nonlinear inflatable actuators for distributed control in soft robots. *Advanced Materials*, page 2301487, 2023.

Comment 3: Another point is with regard to the choice of control variables. In Fig. 1c, the flow curve gives a unique flow rate for an imposed pressure drop, but if the flow rate is fixed, then the pressure drop exhibits multistability. So depending on what the control variable is, the system may or may not exhibit memory, hysteresis, multistability etc.

Response: This is another great comment. Indeed, if one controls flow rate, one isolated valve will display multistability and hysteresis. However, in this work we were interested in understanding how instabilities and memory effects could emerge as a collective phenomenon for valves that do not present such behavior by themselves. On the other hand, in our experiments it is easier to control pressure rather than flow rate. For these reasons we decided to study our valves under pressure boundary conditions, although in the future we will probably also study flow-controlled boundary conditions.

Clarifications:

I think a better explanation and clarification of all these points would be very useful.

Clarification 1: Fig. 3: how does the domain wall in the valve opening propagate? Presumably its speed is set by the ramping of P ? Do you ever see spontaneous drifting of the domain wall for strong enough static pressure drops, i.e., is there an analogue of a trigger (pushed) wave in the system?

Response: This is another very interesting comment, it is actually something that interests us very much. Indeed, in the case presented in this work we study stationary solutions and the position of the “domain wall” is controlled by the external pressure imposed at the boundaries. As the external pressure increases the valves change from one stable branch to another one by one, moving the domain wall one position at a time. This is the phenomena studied in this work: memory effects where stationary solutions are controlled by the external pressure. However, Referee #1 makes a very good point, there is another regime (not studied in this work) where travelling waves appear under constant pressure

boundary conditions. We studied this regime in a previous work but only in the phenomenological model [11]. In a nutshell, it is the value of α what controls if the system display travelling waves. We are working to try to test this theoretical prediction experimentally in a future work.

[11] M. Ruiz-García and E. Katifori. Emergent dynamics in excitable flow systems. *Physical Review E*, 103(6):062301, 2021.

Clarification 2: The model also includes a capacitive element to account for local fluid storage. It was unclear to me what aspects of the phenomenology rely strongly on this feature, and what do not. Perhaps this can be explained a bit more.

Response: We believe that our answer to the first comment can also help with this clarification. Indeed, we show how in this minimalist model there is no instability if we do not let the system accumulate volume. Equivalently, in the system with many NDR valves connected in series, hysteresis and memory effects depend on the capacity to store volume, this can be understood in the following way: if the system cannot accumulate volume anywhere, mass conservation imposes that the flow rate in all the valves is the same at all times, making the system behave as if there was only one valve.

Clarification 3: In relating the single NDR element (Fig. 2a) to the collective (Fig.2b), is there a simple relation between the slopes of the response curves for the network in terms of the individual element flow properties?

Response: Indeed, there is a simple relation between the slope of the inner hysteresis loops and the slopes displayed by the single valve. Following also the advice of Referee #4 we have moved the explanation of these details to the Methods. In particular, the explanation of the relation between the slopes reads now:

“A similar mechanism explains the different slopes for the stable branches in the nested hysteresis loops. We have a system of multiple valves connected in series, and since the valves are always in one of their two PDR branches the effective resistance of the complete system is the sum of the individual resistances. With the peculiarity that now the valves can be in one of their two PDR branches (that have different resistances). Following this idea, the slopes of the left and right sides of the loop in Fig.2b are directly proportional to the slopes of both PDR branches in Fig.2a. However, the slope of the inner hysteresis branches (dashed lines in Fig.2b) have slopes that interpolate between both limit values, depending on how many resistors are in each PDR branch.”

Clarification 4: Lines 399-401: "One plausible explanation for this difference stems from the fact that the 2D valves are composed of two compressible elastic rods that change their geometry due to the absolute pressure, thereby affecting their response to ΔP ." What happens if you keep ΔP the same and increase both the inlet and outlet pressure? This would be a simple check for this hypothesis.

Response: Thank you very much, this comment combined with the main question posed by Referee #4 has led us to create a new section in the supplementary information that we think will be very useful for the reader. We have performed new realistic simulations showing that indeed the flow response to pressure difference changes with the absolute pressure, and we have verified that this improves the agreement between the phenomenological model and the realistic simulations in the system with multiple valves. We reproduce the complete discussion in our answer to Referee #4, but to facilitate the review of our work, this is the main result:

a Simulations with a soft valve presenting the flow versus pressure difference as the pressure at the inlet and outlet are shifted an equal amount P . **b** Comparison between realistic simulation (solid line) and phenomenological model (dashed line) using $\Gamma(\Delta P, P)$ for the system with 8 valves presented in Fig.3 of the main text.

Minor comments

Minor comment 1: Eq. 3: The graph Laplacian is currently unscreened corresponding to a Poisson equation. If one includes screening (so we get a Helmholtz-like equation), then there should be direct dependence on pressure itself. Can this explain the tilted response curves?

Response: Thank you very much for the suggestion. Since we found answering to the previous comment that the disagreement was due to our assumption that the flow through a single valve did not depend on the absolute pressure, we have not carried out this calculation. However we may test this in future works.

Minor comment 2: Eq. 7-8: What is e_θ ? Is the numerics performed in 2D or 3D? The longitudinal coordinate used in main text is x while the coordinate used in Method–simulations is a mixing of r (such as fluid velocity field $v(\mathbf{r}, t)$ and pressure $p(\mathbf{r}, t)$) and x (such as Eqs.7 and 8, and the displacement field of elastic materials). Also, both G and μ_s are used to denote the shear modulus. It would be useful if the notation were made common everywhere and explained more clearly.

Response: We apologize for these typographical errors and are grateful to Referee #1 for catching them. All the fluid-structure numerical simulation are 2D, thus e_θ now reads e_x , where e_x is the stream-wise unit vector. Moreover, to avoid confusion, we have renamed the position vector as \mathbf{x} instead of \mathbf{r} . Finally, we have made sure that we only use the symbol G to denote the elastic shear modulus throughout the manuscript.

Minor comment 3: Eq. 12: Shouldn't the boundary condition be continuity of the normal stress, so $\boldsymbol{\sigma} \cdot \mathbf{n} = \mathbf{T} \cdot \mathbf{n}$?

Response: We thank Referee #1 for raising this comment. However, the way the fluid stress

tensor is defined, $\mathcal{T} = -pI + \nabla\mathbf{v} + \nabla\mathbf{v}^T$, implies that the stress balance at the contact interface between the fluid and the solid is written as $\boldsymbol{\sigma}\cdot\mathbf{n} = -\mathcal{T}\cdot\mathbf{n}$, so that a positive pressure *pushes* the elastic solid.

Reviewer #3:

This paper explores fluid flow networks that contain elements with a non-linear pressure-flow relationship, which is commonly observed in both biological and artificial systems. Through the integration of experimental data, simple theory, and numerical simulations, the authors provide an insight into the collective behavior of such networks. It is intriguing to consider the idea of a fluidic memristors, but the term is not entirely new (see Xiong et al. 2023), and the authors did not demonstrate many possible capabilities (e.g. calculating using such systems, or determining whether biological systems perform calculations). The paper is clear and well written (there were no typographical errors or problems with the presentation of the data). While the authors barely show the potential of this idea, the idea presented is very interesting, and I am in favor of accepting the work in its current form.

Response: We thank Referee #3 very much for recommending our paper for publication without further review. We agree with Referee #3 that it is unfortunate that a recent work also uses the term ‘fluidic memristor’ for a completely different system [12]. In our opinion, this other work study a electric memristor in a fluidic system, in other words, they do not study the flow response to pressure difference as we do, they rather study electric current in response to electric field. The system is actually so different to ours that we have decided not to cite their work in our paper to avoid confusing the reader. Following the advice of Referee #3 we have modified the paper to try to better convey the potential impact of our work.

[12] Xiong, T., Li, C., He, X., Xie, B., Zong, J., Jiang, Y., ... & Mao, L. Neuromorphic functions with a polyelectrolyte-confined fluidic memristor. *Science*, 379(6628), 156-161, 2023.

Reviewer #4:

The authors present an interesting and timely study on a "fluidic memristor", inspired by similar non-linear hysteretic memory-capable elements from electric circuits and beyond, but realised for the first time in the fluid flow network analogy. They combine theory, simulation, and experiments. They explore how this memristor operates, where the hysteresis originates from, and the coupling to elasticity in the chamber and valve elements. Ultimately, and unfortunately, though, the analytic model is illuminating for the effect but are a rather poor match to the simulation and especially the experiment, capturing the real behaviour only in the most qualitative, broad strokes. Finally, with a handful of exceptions detailed later, the paper is well-written, well-motivated, and easy to follow.

Taken all together, it is the opinion of this referee that the manuscript is promising enough to warrant eventual publication in Nature Communications, but requires further polish as well as at least one critical further step in the modelling approaches first.

We thank Referee #4 for their time and effort reviewing our work. In particular, we are very thankful for encouraging us to demonstrate that the phenomenological model can display a much better match with realistic simulations and experiments if one includes the dependence of the flow of a single valve on the absolute pressure (not only on the pressure difference). This has allowed us to write a new section of the supplementary information that we think will be very interesting for the future reader.

The critical point first:

Other than the sawtooth-like behaviour of the flow rate as a function of the pressure drop, nothing about the 1d model matches the experiment, nor does it match particularly well to the 2d simulation. The authors even make a (strong!) speculation about a major reason for these mismatches – the absolute pressure level is simply a fixed boundary condition in the model. However, it would be both very relevant and should also be relatively easy to either make the absolute pressure level either a fitting parameter or at least a parameter to explore further. Given its claimed (and plausible) importance to rectifying the one glaring issue in the paper, this needs to be better addressed, at least in the 1d model. To be clear, the authors could also vary the absolute pressure in the simulation and/or the experiment and that would certainly also add value to the paper, but I think the phenomenological model would probably be enough.

We thank Referee #4 for raising this question. We realize now that we should have not only stated how the phenomenological model could be modified to reach a better match between the model and the realistic simulations or experiments. Instead, we should have shown results validating this speculation. We have carried out the modification that we suggested to the model and we show now in a new section of the supplementary information that indeed this change leads to a much better match between the theoretical model and the realistic simulations.

To facilitate the review of our work we reproduce here the new section containing these new results:

“In the paper, there is a good qualitative agreement between the 1D model and the realistic simulations or the experiments: the model explains all the fundamental mechanisms behind the nested hysteresis loops, the slopes of the branches, etc. However, there is a quantitative difference, the upper and bottom branches of the hysteresis loop in the realistic simulations and experiments are tilted upwards, whereas they are horizontal in the 1D model.

In the 1D model we assume that the flow through the soft valves only depends on the pressure difference across the valve, in other words $\Gamma(\Delta P)$ does not depend on P (the absolute pressure at the entrance of

the valve). We test here this assumption carrying out simulations with the same soft valve that we used in Fig. 3 of the main text. We use only one valve here and we impose an inlet pressure $\Delta P + P$ and an outlet pressure of P , where $P = 0, 0.1, 0.2, 0.3, 0.4, 0.5$. The results of these simulations are presented in the figure below, they show how as P increases the flow is shifted upwards, what can explain the tilting of the hysteresis loop. To incorporate this effect into the 1D model one could use a function Γ that not only depends on ΔP but also on the absolute pressure P .

Note that the curves in the figure below are not multiplied by a simple factor that depends on P , for example, the flow at the local maximum increases by a factor of 2 when P goes from 0 to 0.5 whereas the flow at the local minimum increases approximately by a factor of 5. To include this effect in a simple way we multiply $\Gamma_{NL}(\Delta P)$ in equation (2) by a linear factor that depends on ΔP and P , leading to a new $\Gamma(\Delta P, P)$:

$$\Gamma(\Delta P, P) = (1 + (C_1 + C_2 \Delta P)P) \Gamma_{NL}(\Delta P) \quad (14)$$

where C_1 and C_2 are constants that are adjusted so that the flow at the local maximum and minimum of $\Gamma(\Delta P, P)$ are multiplied by a factor 2 and 5, respectively, when P goes from 0 to 0.5, leading to $C_1 = -0.8$ and $C_2 = 187$.

We now use this $\Gamma(\Delta P, P)$ in the phenomenological model and carry out a simulation with the same system as in Fig. 3 of the main text:

This figure shows how with only this simple expression for $\Gamma(\Delta P, P)$ the phenomenological model agrees quantitatively with the realistic simulation. The two panels present: **a** Simulations with a soft valve presenting the flow versus pressure difference as the pressure at the inlet and outlet are shifted an equal amount P . **b** Comparison between realistic simulation (solid line) and phenomenological model (dashed line) using $\Gamma(\Delta P, P)$, for a system with 8 valves.”

Smaller comments:

Smaller comment 1: The breakdown of the protocol in figure 2 that begins on line 265 is too detailed, too confusing, and too much like a figure caption. Also, the "black solid curve in panel b" referred to in lines 268-9 does not appear to exist?

Response: Thank you very much for the suggestion. We have decided to move the details of this section to the Methods, so that the interested reader can delve into the intricacies of the phenomenon but that it does not break the flow of the story. Indeed there was a typo and lines 268-9 should have

read “black solid curve in panel a”. Nevertheless, since previous Fig.2a also led to some confusion to Referee #1, we have decided to include the panel with the arrows in the Methods. To facilitate the review of our manuscript, this is the new section of the Methods:

“

Flow rate Q as a function of pressure difference ΔP for a single NDR resistor. Points and arrows explain the process that leads to the nested hysteresis loops presented in Fig.2 for a 1D network of NDR resistors. NDR resistors swap one by one from one PDR branch to another as the total pressure imposed to the system (P_{inlet}) increases and then decreases.

To understand the mechanisms underlying the phenomena that leads to the nested hysteresis loops, let us divide the domain of the nonlinear function in Fig.2a into three parts that we will denote as: first positive differential resistance (PDR) branch (from $\Delta P = 0$ to the local maximum), NDR branch (negative slope region), and second PDR branch (from the local minimum onward).

In the protocol displayed by the inset of Fig.2b, we start by increasing P_{inlet} from zero, represented by the bottom red dot in panel a and the red dots at the bottom of panel c (where each point indicates the pressure at each node of the network). As we increase P_{inlet} , all the nonlinear resistors of the network follow the black solid curve in the figure above, represented by the red dots on the first PDR branch of the curve, until now the pressure distribution is homogeneous inside the network. However, once all the resistors reach the local maximum of the figure above, if P_{inlet} keeps increasing, a homogeneous pressure drop along the network becomes unstable. Beyond this pressure, the system finds a different stable solution by swapping one resistor to the second PDR branch. Through this mechanism, as P_{inlet} increases, one by one the nonlinear resistors jump to the second PDR branch, and the system divides into two distinct domains characterized by a high and a low pressure drop, as shown by Fig.2c (red curves). Once all the resistors have swapped to the second PDR branch, the system displays an internal pressure drop that is homogeneous, as shown by the blue upper line in Fig.2c. When the inlet pressure P_{inlet} is decreased following the ramping-down protocol in the inset to Fig.2b, represented by blue dots in the figure above, all the resistors stay on the second PDR branch until they reach the local minimum. Below such P_{inlet} , now one by one resistors jump to the first PDR branch, again forming two pressure drop domains (bottom blue curve in Fig.2c).

All the intricacies of the multiple hysteresis loops present in panel Fig.2b can be understood using similar arguments to the ones used here. The jumps in the upper branch of the hysteresis loop are present because an infinitesimal increase in P_{inlet} leads to a resistor jumping from the first to the second PDR branch. Since the pressure drop is externally controlled for the complete system (P_{inlet}), the pressure

drop across the other NDR valves have to decrease to accommodate this sudden change, leading to the jumps present in Fig.2b every time one NDR valve changes branch. A similar mechanism explains the different slopes for the stable branches in the nested hysteresis loops. We have a system of multiple valves connected in series, and since the valves are always in one of their two PDR branches the effective resistance of the complete system is the sum of the individual resistances. With the peculiarity that now the valves can be in one of their two PDR branches (that have different resistances). Following this idea, the slopes of the left and right sides of the loop in Fig.2b are directly proportional to the slopes of both PDR branches in Fig.2a. However, the slope of the inner hysteresis branches (dashed lines in Fig.2b) have slopes that interpolate between both limit values, depending on how many resistors are in each PDR branch. Finally, it is important to mention that the shape and properties of the multiple hysteresis loops shown in Fig.2b can be directly controlled changing the shape of the nonlinear curve in Fig.2a, and also by changing the number of resistors in the network.”

Smaller comment 2: On line 322 "dots" in Fig. 3b are referred to and it is unclear what is meant by this.

Response: Indeed this was a typo, it should have read Fig. 3e. It is now corrected in the paper, thank you very much for pointing this out.

Smaller comment 3: The authors claim that there are two domains of behaviour "distinctly visible" in Fig. 3b. This is not so clear, and even it were clear, in order to better establish these two distinct domains, the authors should include some form of actual quantification to back up their assertion.

Response: In this case we must admit that the visualization of the two domains was so clear to us that we did not anticipate this could be confusing for the reader. We thank Referee #4 for pointing this out. We have changed this paragraph to improve its clarity. It now reads: “As predicted by our 1D model, the hysteresis cycle is connected to the creation of two domains of low and high pressure drop, corresponding to the two slopes present in the lines shown in Fig.3c and d. These two domains are produced by two groups of valves, each one operating in a different positive differential resistance branch. The upper line in Fig.3d represents the pressure along the middle line of Fig.3b, where the two domains are distinctly visible: the four valves which are closer to the inlet present almost no deformation (low pressure drop), while the other four valves are highly deformed (high pressure drop).”

Smaller comment 4: Generally, throughout all the figures, the authors use symbols, features, and line widths that are too small on a printed page. These should be enlarged to improve readability.

Response: We have enlarged fonts and symbols in figures 2, 3, 4 and 5 of the new manuscript. Thank you very much for the suggestion.

Smaller comment 5: On line 481, in the discussion, the authors say "until now, this phenomenology had been elusive in the fluidic realm" which, in light of the examples of neurovasculature and gymnosperms discussed earlier in the paper and studied in detail elsewhere, I find to be a bit too strong of a statement.

Response: We understand the point. We have rephrased the sentence as: “Until now, this phenomenology had not been carefully analyzed in the fluidic realm.”

Smaller comment 6: In the methods section the authors seemingly swap notation from Q_{ij} to I_{ij} beginning in equation 2 with no warning or explanation.

Response: Thank you very much, indeed this was a typo. We have now fixed it throughout the

Methods.

Smaller comment 7: Throughout page 14 and 15, the usage of G_r vs G is confusing and unclear. Also, it seems likely that G appears entirely mistakenly on line 646? That should likely be μ_s instead.

Response: We apologize for the confusion in the notation. We have fixed the problem by using the subscripts "r" and "m" to denote quantities of the elastic rods and the outer elastic matrix, respectively. The symbol μ is used for the fluid's viscosity only, and G is used to denote the shear modulus of the elastic elements.

Smaller comment 8: On line 632 the authors assert they can set the reference pressure to zero without loss of generality. This should be true in a linear system, but seems highly non-trivial here. The authors should either explain further or amend.

Response: We appreciate the reviewer for bringing up this issue. We agree with the reviewer that the reference pressure at the outlet can indeed play a significant role in a nonlinear system like ours. Therefore, stating that we set $p = 0$ without loss of generality is not accurate. We have revised the wording in that paragraph to address this issue. In fact, we conducted new simulations for a single NDR valve with various fixed outlet pressures, finding that the characteristic Q - ΔP curve changes quantitatively (though not qualitative).

REVIEWERS' COMMENTS

Reviewer #1 (Remarks to the Author):

The authors have satisfactorily addressed the concerns raised by the reviewers.

I happily recommend publication.

Two very minor comments -

Line 686: Maybe the authors can add an arrow in the schematic figure 3a to show which direction the normal points (into the liquid or solid), to avoid confusion?

Line 680, Eq(11): Clarify if and whether geometric nonlinearity in the strain is important. My impression is that it is not.

Reviewer #2 (Remarks to the Author):

Reviewer #3 (Remarks to the Author):

The authors addressed my comments and I recommend acceptance of this work to Nature Communications.